# Relevance of *CYP2D6* Gene Variants in Population Genetic Differentiation

**DOI:** 10.3390/pharmaceutics14112481

**Published:** 2022-11-16

**Authors:** Anita Stojanović Marković, Matea Zajc Petranović, Tatjana Škarić-Jurić, Željka Celinšćak, Maja Šetinc, Željka Tomas, Marijana Peričić Salihović

**Affiliations:** 1Institute for Anthropological Research, 10000 Zagreb, Croatia; 2Department for Translational Medicine, Srebrnjak Children’s Hospital, 10000 Zagreb, Croatia

**Keywords:** *CYP2D6* gene, polymorphism, haplotype, star allele, pharmacogenetics, 1000 Genomes populations

## Abstract

A significant portion of the variability in complex features, such as drug response, is likely caused by human genetic diversity. One of the highly polymorphic pharmacogenes is *CYP2D6*, encoding an enzyme involved in the metabolism of about 25% of commonly prescribed drugs. In a directed search of the 1000 Genomes Phase III variation data, 86 single nucleotide polymorphisms (SNPs) in the *CYP2D6* gene were extracted from the genotypes of 2504 individuals from 26 populations, and then used to reconstruct haplotypes. Analyses were performed using Haploview, Phase, and Arlequin softwares. Haplotype and nucleotide diversity were high in all populations, but highest in populations of African ancestry. Pairwise F_ST_ showed significant results for eleven SNPs, six of which were characteristic of African populations, while four SNPs were most common in East Asian populations. A principal component analysis of *CYP2D6* haplotypes showed that African populations form one cluster, Asian populations form another cluster with East and South Asian populations separated, while European populations form the third cluster. Linkage disequilibrium showed that all African populations have three or more haplotype blocks within the *CYP2D6* gene, while other world populations have one, except for Chinese Dai and Punjabi in Pakistan populations, which have two.

## 1. Introduction

Human CYP2D6 protein was purified in 1984 [1], and the gene was mapped to chromosome 22q13 in 1987 [2]. Two years later, the *CYP2D6* gene was cloned and sequenced [3], and it was discovered that the gene locus contains two additional genes: a non-functional *CYP2D7* gene, and a *CYP2D8* pseudogene. The *CYP2D6* gene contains nine exons and is highly polymorphic.

In 1996, a group of international experts in pharmacogenetics decided to systematize allelic variants of the *CYP2D6* by proposing a haplotype-based star (*) nomenclature system [4]. Since then, more than 135 *CYP2D6* star alleles have been described and are available on the Pharmacogene Variation (PharmVar) Consortium website [5,6].

Genetic variations of the *CYP2D6* affect the metabolizing activity of the CYP2D6 enzyme. Its activity can vary from complete absence to increased activity, and can be grouped into four different drug-metabolism phenotypes: (1) poor metabolizer (PM—two null activity alleles); (2) intermediate metabolizer (IM—one normal activity allele with one null activity allele; or two reduced activity alleles); (3) extensive metabolizer (EM—two normal activity alleles; or a combination of one increased activity allele with one allele of reduced activity); and (4) ultra-rapid metabolizer (UM—one normal activity allele with one increased activity allele) [7,8,9,10]. CYP2D6 is expressed in the human liver where it accounts for only 2–4% of the total CYP content [11,12], but it is involved in the metabolism of up to 25% of drugs commonly used in medicine, including antidepressants, a number of atypical and typical antipsychotics, antineoplastic agents (e.g., tamoxifen), adrenergic antagonists (e.g., metoprolol), and analgesics (e.g., codeine and tramadol) [13,14,15,16,17,18,19]. Variations in the *CYP2D6* gene have also been studied as a risk factor for a number of diseases: Parkinson’s disease [20,21,22], schizophrenia and other psychiatric diseases [16,23], Alzheimer’s disease [24,25], as well as several forms of cancer [26,27].

Although CYP2D6′s role in the metabolism of naturally occurring xenobiotics has not been researched extensively, it is well-known that this enzyme has a very high affinity for alkaloids [28]. Therefore, alkaloid metabolism in food is assumed to have played a role in its evolution. There is a theory that 10,000 to 20,000 years ago in Northwest Africa, due to food shortages compared to population size, the number of plants that could provide usable food increased as a result of selection that favored the survival of individuals capable of more effective detoxification of plant toxins [29]. The best example of how dietary modifications throughout human history have provoked selection pressure on the genes whose products metabolize food molecules is N-acetyl-transferase 2 (e.g., [30]). The current patterns of *CYP2D6* genetic diversity, according to Fuselli (2010), are a result of the selective pressure of new or more potent CYP2D6 substrates that emerged as food choices, particularly at the start of the Neolithic transition, in response to worsening nutritional conditions and higher disease burdens [31].

More genetic variation can be seen in genes encoding detoxification enzymes, which shows that exposure to various substrates also aided in the evolution of genetic variants. Detoxification enzymes really exhibit signs of positive selection, such as modifications of the amino acid sequence that increase substrate selectivity [6,32]. Compared to any other category of pharmacogenomically relevant genes in humans, a recent study revealed that *CYP* genes that metabolize exogenous compounds have far higher frequencies of SNPs that vary greatly between populations [33].

It has not yet been possible to pinpoint the selective factor that causes diet-related patterns of evolution in the *CYP2D6* gene (such as the presence or absence of a particular substrate or a variable concentration of substrates) [34].

Although there are numerous papers on the world distribution of *CYP2D6* variation, most focus on the pharmacological consequences of different variants. The goal of this paper was to identify which single nucleotide polymorphisms (SNPs) and haplotypes in the *CYP2D6* gene determine the genetic specificity of 26 world populations, and to test intra- and inter-group differences in continental groups defined by ancestry. Furthermore, we investigated the role of population differentiation in the definition of the *CYP2D6* star alleles.

## 2. Materials and Methods

The investigated pharmacogene *CYP2D6* is located on the reverse strand of chromosome 22:42,126,499–42,130,865 (GRCh38). Using Data Slicer, a tool implemented on the Ensembl website [35], data on 2504 individuals belonging to 26 world populations from Phase 3 of the 1000 Genomes Project were extracted from Ensembl Release 107 [36]. The data file contained 279 polymorphic positions: 9 insertions/deletions (indels) and 270 single nucleotide polymorphisms (SNPs). Two positions were monomorphic, and in 182 positions the minor allele was found less than five times in the total sample. All indels, monomorphic SNPs, and SNPs where the minor allele occurred less than five times in the total sample were excluded from further analyses, leaving 86 SNPs.

Allele frequencies and the Hardy–Weinberg equilibrium were calculated separately for each population using VCFtools [37]. VCF files were also used to create ped files for linkage disequilibrium (LD) calculation and visualization, which was performed in Haploview software [38]. Haplotype blocks were constructed using the confidence intervals algorithm [39], and the informativeness of the block was further estimated by r^2^ measurement of LD between SNPs defining the ends of haplotype blocks, both implemented in Haploview software. The *CYP2D6* haplotypes were inferred using Phase ver. 2.1 [40,41]. Haplotype frequencies were used for principal component analysis (PCA), performed using the statistical package SPSS Statistics 21.0 for Windows (SPSS Inc., Chicago, IL, USA). The most common haplotypes were translated into the star allele nomenclature using data on the PharmVar website [5].

Indices of intrapopulation molecular diversity (number of haplotypes, polymorphic sites, transitions and transversions) and AMOVA, the statistical significance of which was assessed by generating 100,000 random samples, were calculated using Arlequin 3.5 software [42].

## 3. Results

In order to capture most of the variability in the *CYP2D6* gene, we investigated 86 SNPs within the gene region (22:42,126,499–42,130,865), whose minor allele frequencies (MAF) were higher than 0.1%. The allele frequencies of studied polymorphic sites in 26 world populations from the 1000 Genomes database are shown in Appendix A. Phased SNPs revealed 232 unique haplotypes.

Several diversity indices were calculated in order to see the intrapopulation variation, and the findings are displayed in Table 1.

African populations had the highest number of polymorphic sites: the most poly-morphisms (55) were found in the population of African ancestry in the Southwest USA, and the least (46) in the Mende population in Sierra Leone. The lowest number of polymorphic sites was found in East Asian populations ranging from 20 in the Japanese population to 30 in the Han Chinese population in Beijing and in the Kinh population in Vietnam. The highest number of haplotypes were found in populations of African ancestry, ranging from 37 in the Gambian population to 31 haplotypes in the Yoruba population from Nigeria. The lowest number of haplotypes of all 26 investigated populations was found in the population of Japan (11), followed by the population of Peru (16) and the British population from England and Scotland (17). Interestingly, although considered a genetically isolated population, Finns did not have the lowest number of haplotypes.

Overall, haplotype diversity and nucleotide diversity were high in all populations, and highest in African populations where haplotype diversity ranged from 0.908 to 0.936. The lowest haplotype diversity was observed in East Asian populations ranging from 0.607 (Vietnam) to 0.717 (Japan). The results of the nucleotide diversity analysis are also very similar: the highest diversity was found in African populations, and the lowest in East Asian populations. According to diversity indices, the Japanese population has the lowest genetic variation.

In order to calculate the level of population differentiation, we performed AMOVA analyses. Populations were joined in five continental groups based on shared common ancestry. Approximately 8% of the variation was due to between-group differences (F_CT_ = 0.077), while the interpopulation variation was 9% (F_ST_ = 0.091). When we examined each continental group separately, we discovered that East Asian populations showed the greatest differentiation (F_ST_ = 0.031), while European populations showed the smallest (F_ST_ = 0.002).

To elucidate which of the 86 *CYP2D6* gene SNPs mostly affected the population differentiation, locus-by-locus AMOVA was conducted. Allelic variations at 11 SNPs (rs75203276, rs59421388, rs61736512, rs16947, rs76327133, rs80262685, rs28371706, rs2267447, rs1065852, rs2004511, and rs1081003) contribute to the inter-population differentiation higher than 10%, with F_ST_ values ranging from 0.103 to 0.366 (Table 2).

Figure 1 shows the distribution of minor alleles in those 11 SNPs in 26 world populations. Six SNPs are characteristic of African populations (rs75203276, rs59421388, rs61736512, rs76327133, rs80262685, and rs28371706), with the population of African ancestry in Southwest USA showing somewhat lower frequencies with values of 3–4% in five SNPs. The remaining five SNPs are present in all world populations, with rs16947 being most common in African populations (42–65%), least common in East Asian populations (13–17%), and occurring in a range of 25–44% in the rest of the world population. The SNPs rs2267447, rs1065852, rs2004511, and rs1081003 are the most common in East Asian populations with frequencies in the range of 60–68%, while in the Japanese population they were in the range of 36–39%. The most common SNPs for non-African and non-Asian populations are rs16947, rs226744, rs106585, and rs200451.

The distribution of haplotypes in all 26 investigated populations is shown in Figure 2. Haplotypes 1 and 2 are present in all world populations with varying frequencies, and their combined frequencies range from a minimum of 16% in the population of Sierra Leone to a maximum of 82% in the Peru population. Haplotype 3 is distributed in all populations except the Finnish population. However, it is most characteristic of East Asian populations, with the lowest distribution of 36% in Japan, while other East Asian populations have it in the range of 56–60%. This haplotype is found in some African populations with a frequency of more than 5% (Nigeria, the African Caribbean and Gambia) and higher than 10% in Sierra Leone and Bangladesh. The distribution frequencies of haplotype 4 are over 10% in all European populations (minimum 12% in Finland, maximum 19% in Great Britain), and in populations of Puerto Rico and Colombia. Haplotype 5 is most common in Southeast Asian populations, but has a frequency distribution of over 10% in Italian and Central European populations. Haplotypes 6, 7, 8, and 10 are mainly characteristic of African populations, where the frequency of distribution is generally higher than 5%. All haplotypes which occurred less than five times were presented together (rest) in Figure 2.

All populations share the most common haplotype determining the star allele *1. The eight most common haplotypes account for 74% of all haplotypes worldwide. Haplotype 11 translated into star allele *10 is typical for East Asian populations. The most common haplotypes in Europe are *2 and *4. In South Asian populations, the most common haplotypes are *2 and *41. African populations are the most specific: star alleles *1, *29, and *17 are predominantly found in these populations.

To compare world populations based on *CYP2D6* haplotypes, we performed a principal component analysis (PCA) (Figure 3). Its results showed that African populations form one cluster, Asian populations another cluster with East and South Asian populations separated, and European populations form a third cluster. South American populations do not have a distinct cluster: Colombian and Puerto Rican populations overlap with European populations, while Mexican and Peruvian populations are closer to Southeast Asian populations.

Linkage disequilibrium (LD) was calculated and visualized using Haploview 4.2 software, which constructs blocks based on D’ values. All African populations have three or more haplotype blocks within the *CYP2D6* gene, while other world populations have one haplotype block, except Chinese Dai and Punjabi in Pakistan populations, which have two. A large block of the Chinese Dai population (in the range rs1135840–rs28371702) has an r^2^ of 0.888 and a large block of the Punjabi population in Pakistan (in the range rs16947–rs1080995) has an r^2^ of 0.976, while both small blocks in these two populations have an r^2^ below 0.8. In all African populations but Yoruba from Nigeria, we observed the *CYP2D6* gene haplotype block ranging from rs1081000 to rs1080995 (r^2^ > 0.8). All African populations except the African ancestry in the southwest USA population share the same haplotype block in the range rs1135840–rs27371730 (with r^2^ substantially below 0.8), and ASW is the only African population to have all its blocks with r^2^ greater than 0.8. The Yoruba in Ibadan (Nigeria) population has a total of four haplotype blocks, but only one block is in complete LD (in the range rs75203276–rs61736512). The same haplotype block occurs in the Mende in Sierra Leone population (r^2^ of 0.935) and the African Caribbean in Barbados population (r^2^ of 1.0). In addition to the block in the range rs1081000–1065850, the Luhya in Kenya population has another one that is in high LD (in the range rs75203276–rs76327133, r^2^ of 0.902). The Gambian in Western Division-Mandinka population’s second haplotype block in high LD is the one in range rs569421388–rs76327133, with an r^2^ of 0.868.

The haplotype block found In the European continental group, ranging from rs1135840 to rs1065852, in all five populations has r^2^ values substantially below 0.8. The same haplotype block was also detected in three East Asian populations (Japanese and two Han Chinese populations), in three South Asian populations (Indian Telugu, Bengali in Bangladesh, and Sri Lankan Tamil in the UK), and in two American populations (Colombian and Puerto Rican), also with r^2^ below 0.8. The only remaining East Asian population, Kinh from Vietnam, has a haplotype block bit shorter than the one previously mentioned, ranging from rs28371730 to rs1065852. In contrast, the population of Gujarati Indians in the USA, the last remaining South Asian population, also has a shorter block ranging from rs1135840 to rs1080995, with both r^2^ below 0.8. The latter block, again with r^2^ < 0.8, was also found in the Mexican Ancestry in Los Angeles and Peruvian populations.

## 4. Discussion

Genetic polymorphisms are responsible for a substantial proportion of inter-individual and inter-ethnic heterogeneity in drug response [43]. A number of studies investigated the distribution of genetic variants responsible for heterogeneity in drug response in different populations [44,45,46]. In this study, 86 polymorphic SNPs within the *CYP2D6* gene were analysed in 26 world populations from the 1000 Genomes database [36], in order to estimate the influence of *CYP2D6* haplotype diversity on population differentiation.

The estimated F_ST_ (0.07) indicates a moderate overall genetic differentiation level. However, when continental population groups were examined separately, European populations showed more than 10 times lower interpopulation differentiaton than East Asian populations, which showed the highest. Similar results were found in a study by Jay and colleagues (2013), which showed that the lowest genetic variation of the ADME genes was in Europe, followed by Asia, Africa, and the Americas. In general, the difference between two European populations separated by 1000 km is far less than in other world populations [47]. Geographical isolation together with various selection forces leads to an increase in F_ST_ values among human populations [48].

Locus-by-locus AMOVA revealed 11 SNPs with F_ST_ values greater than 10%. After calculating the MAFs of these 11 SNPs, we observed a clear clustering of SNPs in relation to the studied populations: four SNPs were present in all world populations, but the most frequent in East Asian populations (rs2267447, rs1065852, rs2004511, and rs1081003), six SNPs were almost exclusively found in African populations (rs75203276, rs59421388, rs61736512, rs76327133, rs80262685, and rs28371706), while rs16947 was found in all world populations.

Four SNPs of the *CYP2D6* gene characteristic of East Asian populations occur in Vietnamese and three Chinese populations with a frequency in the range of 60–68%, and the Japanese population in the range of 36–39%. The finding that the Japanese population has frequencies that deviate from the rest of the group is not surprising since Japan regularly diverges due to their relative isolation throughout history. The rs1081003, a synonymous variant (Phe112Phe), showed the highest overall F_ST_ value of 0.37, and due to its substantially higher frequency in East Asian populations, it can be considered a typical East Asian variant. In addition, this SNP’s minor A allele was also found to be the major allele in some South East Asian populations [49]. The Pharmacogene Variation (PharmVar) Consortium database defines rs1081003 as a suballele of numerous star alleles (*2, *4, *10, *36, *37, etc.).

SNP rs2004511, an intron variant with an overall F_ST_ value of 0.21, is also predominant in East Asian populations. In 2018, the association of a minor C allele with response to Tramadol was recorded in the ClinVar archive. The PharmVar defines that this SNP determines suballeles for a number of star alleles (*4, *10, *36, *39, etc.).

SNP rs1065852 in East Asian populations (except Japan) has frequencies of its A allele over 60%, making the G allele a minor allele in these populations. In addition to the four East Asian populations represented in the 1000 Genomes, its high frequency was also noticed in populations of Lisu [50] and Wa, both from Yunnan Province of China [51]. According to the PharmGKB, the A allele which causes a missense variant is associated with decreased clearance of alpha-hydroxymetoprolol in healthy individuals compared to the G allele. It is also associated with S-didesmethyl-citalopram plasma concentrations when treated with citalopram or escitalopram in people with depressive disorder. The GG genotype of rs1065852 is associated with a prolonged QTc interval when treating individuals with schizophrenia with iloperidone. Lee et al. (2016), analyzing the association between CYP polymorphisms and blood concentrations of hydroxychloroquine (HCQ) and its metabolite N-desethyl HCQ (DHCQ) in Korean patients with lupus, observed that patients with the GG genotype of allele *10 had the highest [DHCQ]/[HCQ] ratio, while patients with genotype AA had the lowest ratio [52]. López-García et al. (2017) found that this SNP, as it is included in the star allele *4, can affect the effectiveness of antiepileptic drugs [53]. Together with the SNP rs1081003, rs1065852 is in high LD in the populations of the Philippines, Thailand, Vietnam and Laos, and mutations in these key SNPs that define the star alleles *10 and *54 cause reduced CYP2D6 enzyme activity [49]. This SNP is part of the core of numerous star alleles (*10, *36, *37, etc.). The fourth SNP typical for East Asia is rs2267447, whose minor C allele causes change in the intron variant and is associated with response to Tramadol. According to the PharmVar, this SNP defines suballeles for numerous star alleles (*4, *10, *36, *39, etc.).

SNPs characteristic for the African group of populations were almost completely absent in other populations. rs28371706, whose minor A allele causes a missense variant, is the core SNP for defining star alleles *17, *40, *58, *64, *82, *141, and *154. The *CYP2D6*17* star allele occurs in at least 30% of Africans [54,55], and is associated with reduced enzyme activity-individuals carrying the *17 allele that are classified as intermediate metabolizers (IM). According to ClinVar, rs28371706 is associated with response to Tamoxifen and Deutetrabenazine.

The remaining five SNPs characteristic for the African populations have very similar MAF frequencies within each population. In PharmVar, SNPs rs80262685 (T > C, intron variant) and rs76327133 (G > A, intron variant) are associated with suballeles *2, *29, *146, *149, *155, *156, and *157, while the intron variant rs75203276 (C > T) is associated with suballeles *29, *155, *156, and *157. SNPs rs61736512 and rs59421388 (both C > T, missense variants) define the core of several star alleles; both define alleles *29, *70, *149, *155, *156, and *157, while rs61736512 also defines allele *107, and rs59421388 defines allele *109. Those two variants are significantly associated with the decreased metabolism of debrisoquine, according to the PharmaGKB.

The distribution of rs16947 is the most intriguing. If we exclude the population of African ancestry from the Southwest USA where MAF was 42%, its MAF ranged from 50–65% in African populations, to 13–17% in East Asian populations. Muaymbo et al. (2022) found that this SNP had a significantly lower MAF among Africans in Southern Africa (12%) compared to their counterparts in West (65%) and East (56%) Africa [56]. The MAF of rs16947 in Southern Africans is lower than in any other world population, but is closest to the frequencies in East Asian populations. The rs16947 mutation (G > A), causing a missense variant which can result in decreased CYP2D6 enzyme activity, is one of the major mutations that distinguish star alleles *10A and *54 [49]. In the ClinVar archive, this SNP is associated with the Tamoxifen response, the Deitetrabenazine response, and the ultrarapid metabolism of Debrisoquine. The PharmVar defines that this SNP determines a number of core alleles.

The distribution of MAFs of the 11 SNP variants distinguishes African and East Asian populations from others. Haplotype-based PCA analyses showed separate clusters of African and East Asian populations, while South Asian, European and American populations were much less separated. Separation of African populations follows their genetic history and is visible in different genetic studies as well [57,58]. The clustering of East Asian populations is also evident from studies of other pharmacogenes. Li et al. demonstrated that ADME genes exhibit distinct patterns of population differentiation in a global and regional context. While some genes are conserved (e.g., *SLC04C1* and *NAT1*), others (e.g., *CYP3A5*) exhibit high levels of world population differentiation. On the other hand, the global diversity of some genes primarily reflects differentiation within a particular geographical region, such as Africa, Europe, or East Asia [59]. The genomic diversity of modern populations reflects former demographic and evolutionary changes. In isolated populations with minimal gene exchange, genetic distinctiveness is especially evident (e.g., Jewish populations, Saami, Roma, Basque), which can also be seen in pharmacogene research [60,61,62,63,64,65,66].

Population variation has been studied through patterns of haplotype blocks. African populations have the most diverse block pattern, while it is the most homogeneous in Europe, followed by the Americas, and South and East Asia. The investigated haplotype blocks, based on confidence intervals [39], encompass the region of SNPs with strong LD as a consequence of lack of recombination. The largest number of blocks present in African populations is consistent with their genetic history. The African population has had a relatively large effective population size over a long period of time, allowing recombination events to leave their mark on the haplotype block structure. The significant correlation between SNPs defining the ends of haplotype blocks further supports their informativeness in the African population.

Haplotypes containing functional/associated variants are more likely to determine clinical drug metabolism phenotypes than a single independent SNP [67]. Among the present investigated haplotypes translated into star alleles nomenclature, haplotype 1 defines *1 allele, which is a normal metabolizer and is the most common haplotype in all populations outside of East Asia, where it is the second most common. Haplotype 2 defines *2 allele and is the second most abundant in European, South Asian, and American populations. Haplotype 3 defines *10 allele and is very characteristic of East Asian populations. Haplotype 4 defines *4 allele and appears in European and American populations. Haplotype 5 defines *41 allele and occurs in South Asian and European populations. Haplotypes 6 (*29), 7 (*1), and 8 (*17) occur only in African populations.

Sistonen and colleagues demonstrated different distributions of the CYP2D6 slow (i.e., *9, *10, *17, *29, *45–46) and null-function (i.e., *4, *5, *6) alleles on different continents, probably caused by demographic events [68]. Decreased function CYP2D6 enzymes are characterized by substrate-dependent catalytic properties (gene variants *10, *17, and *29) and enzyme inhibitor affinities (*10, *17), which contribute to a broader spectrum of metabolic responses [69,70,71]. Individuals defined as poor (slow) metabolizers may metabolize certain classes of chemical compounds better (or worse), which should not necessarily be unfavorable. For example, if toxic compounds were activated through CYP2D6-mediated metabolism, slow metabolizers could be at reduced risk of adverse effects [31]. Many widely used therapeutic medications, drugs of abuse, exogenous chemicals such as alkaloids, herbicides, and some endogenous molecules such as progesterone, estrogen, and many other substances are substrates for the human CYP2D6 enzyme [72].

## 5. Conclusions

African populations showed the highest variability of the *CYP2D6* haplotypes, which is consistent with all known studies of genetic variability in humans. However, the greatest differentiation is found among East Asian populations due to extreme homogeneity of the Japanese population and the specific distribution of haplotypes among the Chinese population. The greatest continental homogeneity is found in Europe, followed by South Asia and the Americas.

Locus-by-locus analyses revealed 11 SNP loci affecting inter-population differentiation, six of which are specific to African, four to East Asian populations, while one is present globally. Five of these SNPs (rs2004511, rs1065852, rs2267447, rs28371706, and rs16947) contribute to the known pharmacogenomic effects on clinical outcomes of drugs metabolized by CYP2D6.

## Figures and Tables

**Figure 1 pharmaceutics-14-02481-f001:**
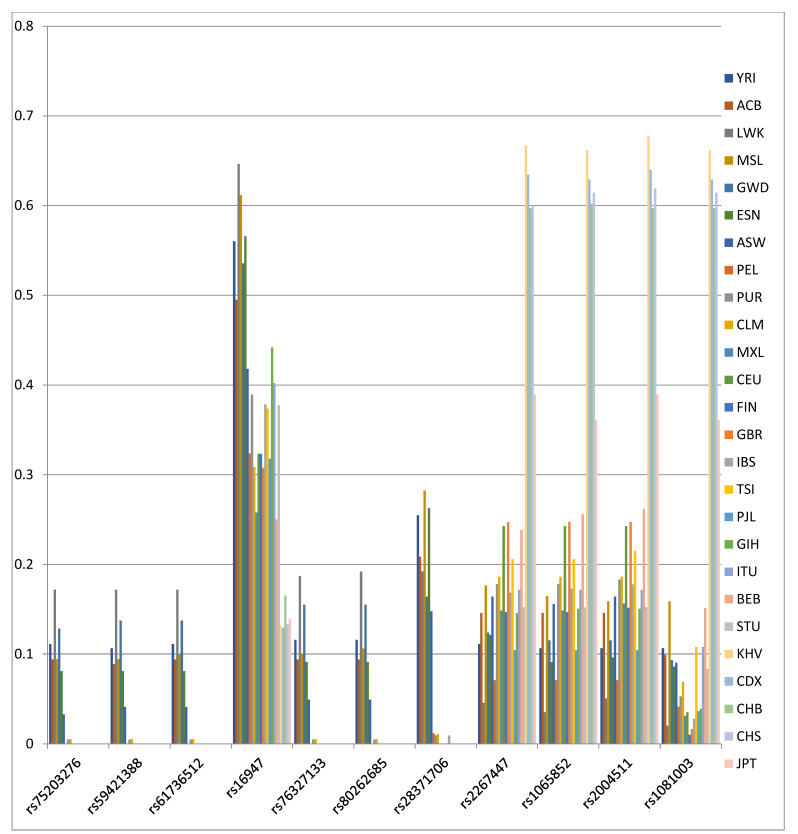
Distribution of minor alleles frequencies in 11 SNPs of the *CYP2D6* gene in 26 world populations.

**Figure 2 pharmaceutics-14-02481-f002:**
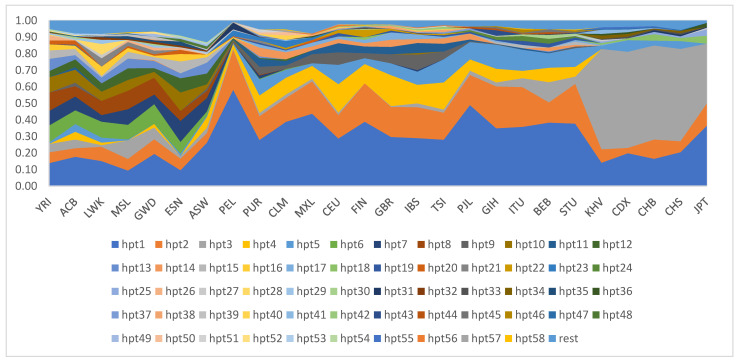
Distribution of the *CYP2D6* gene haplotypes in 26 world populations.

**Figure 3 pharmaceutics-14-02481-f003:**
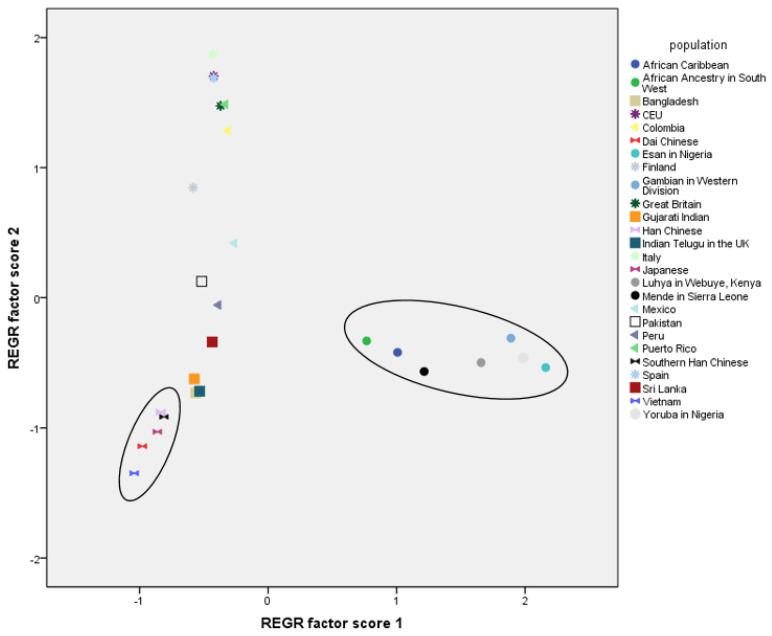
Principal component analysis (PCA) based on *CYP2D6* haplotypes present in 26 world populations.

**Table 1 pharmaceutics-14-02481-t001:** The diversity indices in 26 world populations from the 1000 Genomes database.

		Sample Size	No. of Haplotypes	No. of Polymorphic Sites	Haplotype Diversity	Nucleotide Diversity
**African** **Ancestry**	**ESN**	198	36	50	0.935	0.113
	**GWD**	226	37	49	0.918	0.114
	**MSL**	170	32	46	0.933	0.113
	**YRI**	216	31	48	0.926	0.109
	**LWK**	198	35	48	0.936	0.113
	**ACB**	192	36	48	0.933	0.115
	**ASW**	122	36	55	0.908	0.109
**American** **Ancestry**	**CLM**	188	26	44	0.813	0.106
	**MXL**	128	23	36	0.764	0.092
	**PEL**	170	16	34	0.602	0.086
	**PUR**	208	32	56	0.875	0.113
**European** **Ancestry**	**CEU**	198	21	38	0.847	0.112
	**FIN**	198	19	36	0.778	0.102
	**GBR**	182	17	35	0.834	0.112
	**IBS**	214	27	47	0.851	0.110
	**TSI**	214	22	41	0.845	0.113
**South Asian Ancestry**	**BEB**	172	22	34	0.809	0.099
	**GIH**	206	22	36	0.787	0.106
	**ITU**	204	21	34	0.794	0.105
	**PJL**	192	18	37	0.713	0.094
	**STU**	204	28	38	0.784	0.102
**East Asian Ancestry**	**CDX**	186	18	26	0.620	0.069
	**CHB**	206	19	30	0.637	0.075
	**CHS**	210	20	27	0.643	0.071
	**JPT**	208	11	23	0.717	0.074
	**KHV**	198	22	30	0.607	0.066

Population abbreviations: European ancestry: CEU (Utah residents with Northern and Western ancestry), FIN (Finland), GBR (British in England and Scotland), IBS (Iberian population in Spain), TSI (Toscani in Italy); South Asian ancestry: BEB (Bengali in Bangladesh), GIH (Gujarati Indian), ITU (Indian Telugu in the UK), PJL (Punjabi in Lahore Pakistan), STU (Sri Lankan Tamil in the UK); African ancestry: ACB (African Caribbean in Barbados), ASW (African Ancestry in South West USA), ESN (Esan in Nigeria), GWD (Gambian in Western Division), LWK (Luhya in Webuye, Kenya), MSL (Mende in Sierra Leone), YRI (Yoruba in Ibadan, Nigeria); American ancestry: CLM (Colombian in Medellin, Colombia), MXL (Mexico), PEL (Peruvian in Lima, Peru), PUR (Puerto Rican in Puerto Rico); East Asian ancestry: CDX (Dai Chinese), CHB (Han Chinese in Beijing), CHS (Southern Han Chinese), JPT (Japanese in Tokyo), KHV (Kinh in Ho Chi Minh City, Vietnam).

**Table 2 pharmaceutics-14-02481-t002:** List of 11 single nucleotide polymorphisms (SNPs) of the *CYP2D6* gene which show inter-population differentiation (F_ST_) above 0.1. F_CT_ defines the proportion of total genetic variability among continental groups.

SNP	Position	F_ST_	*p*	F_CT_	*p*
rs1081003	42129754	0.366	<0.00001	0.343	0.00001
rs2004511	42127209	0.211	<0.00001	0.193	0.00001
rs1065852	42130692	0.209	<0.00001	0.188	<0.00001
rs2267447	42128694	0.204	<0.00001	0.187	<0.00001
rs28371706	42129770	0.197	<0.00001	0.163	0.0001
rs80262685	42128576	0.117	<0.00001	0.096	0.00015
rs76327133	42128668	0.115	<0.00001	0.094	0.00025
rs16947	42127941	0.111	<0.00001	0.099	<0.00001
rs61736512	42129132	0.105	<0.00001	0.085	0.00026
rs59421388	42127608	0.104	<0.00001	0.084	0.0002
rs75203276	42128499	0.103	<0.00001	0.081	0.00014

## Data Availability

All data analyzed in this study are available at http://roma.inantro.hr/en/, (accessed on 5 October 2022). In the case of using this database for further analyses, please cite this publication. If further clarification is required, contact the corresponding author.

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
