# Peer review of "Relevance of CYP2D6 Gene Variants in Population Genetic Differentiation"

_pharmaceutics, 2022, doi:10.3390/pharmaceutics14112481_

Round 1

Reviewer 1 Report

Marković and colleagues reconstructed haplotypes of the CYP2D6 gene in various populations.

The subject matter is of interest. 

Specific comments include:

* The English write-up is sometimes awkward and needs editing throughout.

* Please specify which version of  the database was used.

* Please focus the Discussion on the eeesntial since it seems overly verbose.

* Consider discussing the function implications for drug metabolism of your findings.

Author Response

Please see the attachemnt.

Reviewer 2 Report

CYP2D6 is a gene that encodes an enzyme involved in the metabolism of some of the commonly prescribed drugs.

There is an article on the world distribution of the CYP2D6 variation.

However, this article found which single nucleotide polymorphisms (SNPs) and haplotypes in the CYP2D6 gene determine the genetic specificity of 26 world populations.

86 single nucleotide polymorphisms (SNPs) in the CYP2D6 gene were derived from the genotypes of 2,504 individuals from 26 populations.

Intercontinental and intercontinental differences were tested. African populations showed the highest variability of the CYP2D6 haplotype. The Japanese population was found to be extremely homogeneous. The greatest homogeneity was found in Europe. Eleven SNP loci showing inter-population differentiation were shown. Six of these are native to Africa and four to East Asian populations.

As a result, with all these data, it is a scientifically original, important epidemiological study that contributes to the literature and has a high citation potential.
